# Response of Native and Non-Native Subarctic Plant Species to Continuous Illumination by Natural and Artificial Light

**DOI:** 10.3390/plants13192742

**Published:** 2024-09-30

**Authors:** Tatjana G. Shibaeva, Elena G. Sherudilo, Alexandra A. Rubaeva, Natalya Yu. Shmakova, Alexander F. Titov

**Affiliations:** 1Institute of Biology, Karelian Research Center, Russian Academy of Sciences, Petrozavodsk 185910, Russia; sherudil@krc.karelia.ru (E.G.S.); arubaeva@krc.karelia.ru (A.A.R.); titov@krc.karelia.ru (A.F.T.); 2Polar-Alpine Botanical Garden, Kola Scientific Center, Russian Academy of Sciences, Kirovsk 184256, Russia; shmanatalya@yandex.ru

**Keywords:** subarctic, native, non-native plants, photoperiod, continuous lighting, natural and artificial light

## Abstract

This study addressed the following questions: How does continuous lighting (CL) impact plant physiology, and photosynthetic and stress responses? Does the impact of CL depend on the source of the light and other environmental factors (natural vs. artificial)? Do responses to CL differ for native and non-native plant species in the subarctic region and, if differences exist, what physiological reasons might they be associated with them? Experiments were conducted with three plants native to the subarctic region *(Geranium sylvaticum* L., *Geum rivale* L., *Potentilla erecta* (L.) Raeusch.) and three non-native plant species *(Geranium himalayense* Klotzsch, *Geum coccineum* Sibth. and Sm., *Potentilla atrosanguinea* Loddiges ex D. Don) introduced in the Polar-Alpine Botanic Garden (KPABG, 67°38′ N). The experimental groups included three species pairs exposed to (1) a natural 16 h photoperiod, (2) natural CL, (3) an artificial 16 h photoperiod and (4) artificial CL. In the natural environment, measurements of physiological and biochemical parameters were carried out at the peak of the polar day (at the end of June), when the plants were illuminated continuously, and in the second week of August, when the day length was about 16 h. Th experiments with artificial lighting were conducted in climate chambers where plants were exposed to 16 h or 24 h photoperiods for two weeks. Other parameters (light intensity, spectrum composition, temperature and air humidity) were held constant. The obtained results have shown that plants lack specific mechanisms of tolerance to CL. The protective responses are non-specific and induced by developing photo-oxidative stress. In climate chambers, under constant environmental conditions artificial CL causes leaf injuries due to oxidative stress, the main cause of which is circadian asynchrony. In nature, plants are not photodamaged during the polar day, as endogenous rhythms are maintained due to daily fluctuations of several environmental factors (light intensity, spectral distribution, temperature and air humidity). The obtained data show that among possible non-specific protective mechanisms, plants use flavonoids to neutralize the excess ROS generated under CL. In local subarctic plants, their photoprotective role is significantly higher than in non-native introduced plant species.

## 1. Introduction

All living organisms are constantly exposed to various environmental factors and their fluctuations. The most vulnerable to their effects are sessile organisms, such as terrestrial plants that are not capable of moving in space. This means that their survival is directly dependent on the reliability and effectiveness of the protective and adaptive mechanisms that they have acquired over the course of long-term evolution. Some of them are general (non-specific), and some are specific. The first ones are the most universal and therefore are used more often by different plants in a wide variety of natural climatic conditions. It is not surprising that many plant species are characterized by significant numbers and large habitat areas, which include territories with significantly different environmental conditions. Since plants are constantly exposed to fluctuating environmental factors, this forces them to activate, depending on the situation, one or another part of their arsenal of adaptive responses. This is convincingly demonstrated by the results of numerous studies on the effects of environmental factors, such as temperature, moisture supply and light, which are considered to be the main ones for plant life. The third one (light) has been given, as it seems to us, somewhat less attention in recent decades than before, but with the advent of LED light sources and plant factories with artificial lighting (PFAL), the interest in this factor has increased significantly.

Three principal characteristics of light affect plant growth: quantity, quality and duration. The photoperiod quantifies the amount of time a plant is exposed to light, usually within a 24 h circadian cycle. The photoperiod exerts profound effects on the growth and flowering of many plant species, and the manipulation of daylength is essential for scheduling several greenhouse crops. Under natural conditions, the duration of daylight varies considerably over the Earth’s surface, and is dependent on season and latitude. The duration of daylight for 67°44′ N (the Kola Peninsula) (Figure 1) is presented in Table 1. At this latitude, the duration of daylength during the vegetation season (May–September) varies from about 18 h in early May to 24 h in June and early July, and 12 h in late September. The polar day, when the sun remains visible at midnight, lasts at these latitudes for about 50 days (from late May to mid-July) (Figure 2). This natural phenomenon occurs in the summer months in places north of the Arctic Circle or south of the Antarctic Circle. Moreover, at these latitudes, the period when civil twilight lasts all night (also referred to as white night) takes place almost a month before the beginning of the polar day and as many after. At this time, the sun does not sink low below the horizon and darkness does not occur. It turns out that plants in this region are exposed to continuous lighting (CL) for more than 3 months.

However, too much light can be harmful to plants as it significantly reduces the efficiency of photosynthesis which can lead to photoinhibition, causing damage to photosystem II (PSII) [1]. Long photoperiods can also be harmful to plants. While the photoperiodic threshold is different for each plant species, generally, photoperiods longer than 17 h can cause leaf damage observed as interveinal chlorosis, and plants grown under CL tend to have higher carbohydrate and reactive oxygen species (ROS) levels which may lead to the downregulation of photosynthesis [2,3]. Theoretically, the implementation of a CL strategy can increase the yield if photoperiod-related injury is prevented [4,5]. Long photoperiods of lower light intensities could be used to achieve the desired daily light integral (DLI) with lower installed light capacity/capital costs and low electricity costs in regions with low night electricity prices [6]. Therefore, the application of CL in controlled environment agriculture (CEA) is a particular issue of interest to both plant scientists and commercial plant producers [7].

The underlying mechanism of CL injury is unknown. Current hypotheses include (a) a mismatch between the endogenous circadian rhythm and exogenous environmental cues [8,9], (b) improper gene expression [10,11], and (c) the over-accumulation of photosynthetic products leading to feedback inhibition [7,12,13]. Without question, growth under CL can cause plants to undergo a stress response. However, the extent to which the abiotic factor causing the stress response becomes harmful to the plant as opposed to beneficial is difficult to quantify [6].

The use of artificial light generates light environments that are usually very different from the natural, sunlight-illuminated environments under which plants originally evolved [8]. Plants growing in natural environments experience diurnal fluctuations in irradiance and light quality, with gradual shifts between light and dark at dawn and dusk [14,15]. In contrast, experiments in controlled environment chambers typically expose plants to a constant irradiance during the day, and abrupt transitions between light and dark at dawn and dusk. Growth chambers do offer some advantages: the experiments are easily reproducible and allow responses to a single factor (e.g., day length, irradiance, or temperature) to be investigated [16,17]. In the strict sense, CL does not naturally occur [3], not even during the summer in arctic regions because, during the polar day, considerable variations in light intensity and spectral distribution take place. However, for simplicity, we will refer to the lighting conditions during the polar day as CL.

In this study, we conducted an experiment to test the effect of CL on the responses of three native and three non-native subarctic plant species (Figure 3). We assumed that native plants have evolved over thousands of years to be adapted to the specific lighting conditions of the northern polar region. Non-native plants were introduced to the Polar-Alpine Botanic Garden (67°38′ N), which is one of the three botanical gardens in the world located beyond the Arctic Circle. They originated from regions where CL conditions never exist and therefore where plants should not have been adapted to CL. Thus, we observed plants that grew under natural CL during the polar day and plants that were placed in controlled environment chambers with a 24 h photoperiod. We asked the following questions: (1) How does CL impact plant physiology, photosynthetic and stress response? (2) Does the impact of CL depend on the source of light and other environmental factors (natural vs. artificial)? (3) Do responses to CL differ for native and non-native plant species in the subarctic region and, if differences exist, what physiological reasons might they be associated with?

## 2. Results

### 2.1. Plants under Artificial Lighting

Plants grown in climate chambers and treated by A-CL had significantly lower values of potential quantum yield of photochemical activity of PSII (*F_v_*/*F_m_*) compared to A-16 h plants (Table 2). The values of *F_v_*/*F_m_* varied between 0.691 and 0.806 for A-CL plants.

Leaf mass per area (LMA) values were much greater in A-CL plants than in their A-16 h counterparts. The LMA values of the A-CL-treated plants were 7–53% higher for native species and 52–66% for non-native species (Table 2).

The total chlorophyll (Chl) content was decreased in A-CL-treated plants by 9–38% in native species and 19–53% in non-native plant species. The carotenoids were hardly affected by the A-CL in native plant species, but their content decreased in non-native *G. coccineum* and *P. atrosanguinea*. The A-CL increased the Chl *a*/*b* ratio and decreased the Chl/carotenoids ratio in plants, although not all the differences were significant. The share of Chl in light-harvesting complex II (LHCII) was decreased by the A-CL treatment in *G. sylvaticum*, *G. rivale* and *P. atrosanguinea*.

All plants grown under A-CL had higher H_2_O_2_ content compared to the A-16 h plants. Thus, it increased by 39–138% in native plants and by 37–68% in non-native plants (Table 2). The intensity of lipid peroxidation was also higher in these plants. The MDA content increased by 27–93% in native species and by 45–88% in non-native plant species. Additionally, they had higher flavonoid and anthocyanin contents. Native and non-native plants had 27–83% and 5–56% higher content of flavonoids, correspondingly. Anthocyanin content showed a broad range of species-specific variations. The A-CL-induced increase in anthocyanins content varied between 18% in *P. erecta* to 5.3- and 8-fold increases in *G. sylvaticum* and *P. atrosanguinea.*

An analysis of the daily course of the stomatal conductance (*g*_s_) of the plants showed that after two weeks of exposure to A-CL under constant temperature and air humidity, the variation range of *g*_s_ significantly decreased compared to that at A-16 h treatment by 2.6–6 times. The maximum and minimum g_s_ values in the daily cycle of A-16 h and A-CL plants are presented in the Table 3. As an example, the daily course of *g*_s_ in *P. atrosanguinea* is shown on Figure 4. The other species demonstrated similar trends. In all species, A-CL made the sinusoidal pattern of the daily course of *g*_s_ to be flatter, i.e., the damping of oscillation was observed.

Exposure of plants to A-CL resulted in leaf injuries of varying degrees. In some plants, such as *P. erecta* and *G. rivale,* the leaves demonstrated just slight degreening. Injuries to leaves of *G. sylvaticum*, *G. himalayense*, *G. coccineum* and *P. atrosanguinea* were seen as yellow discoloration with tissue necrosis along the margins. Some of the leaves of *G. sylvaticum* and *G. coccineum* have turned rusty red (Figure 5).

### 2.2. Plants under Natural Lighting

The plants exposed to N-CL during the polar day did not have any visible leaf injuries. Much later, in August, when days were shorter, we observed age-dependent injuries that were similar to those seen on the A-CL-treated leaves (Figure 6).

During the polar day, the N-CL plants had significantly similar or slightly lower values of *F_v_*/*F_m_* compared to the N-16 h plants (Table 4). However, the lowest recorded value of *F_v_*/*F_m_* was 0.794, which means that the PSII rection centers were not damaged and N-CL-induced photoinhibition did not occur either in native or non-native plants.

Leaf mass per area (LMA) values were higher by 7–52% and 12–56% in N-CL-treated native and non-native plants, correspondingly, than in their N-16 h counterparts (Table 4).

All N-CL plants showed a decreased in Chl content to varying degrees, although the most pronounced decrease was observed in non-native plant species (Table 4). A N-CL-induced increase in Chl *a*/*b* and a decrease in the share of Chl in LHCII were recorded for all plant species. The carotenoid content was decreased by N-CL by 15–26% in native plant species and by 40–86% in non-native plants. The ratio Chl/carotenoids was hardly affected by N-CL in native plants, but significantly decreased in the non-native plants.

All the plants grown under N-CL had higher H_2_O_2_ content compared to A-16 h plants. It increased by 22–47% in native plants and by 3–90% in non-native plants (Table 4). The MDA content increased by 7–22% in native and by 6–22% in non-native plant species.

Flavonoid content changes in response to N-CL significantly varied between plants in a species-specific manner. For instance, N-CL reduced their content by 20% in *G. hymalayense*, but increased by 70% in *P. erecta* (Table 4).

Under natural conditions, the daily course of stomatal activity did not differ significantly between N-16 h and N-CL plants and the difference between maximum and minimum *g*_s_ values in the daily cycle was only 2–12% smaller in N-CL plants (Table 3). It means that under polar day conditions natural CL did not make the sinusoidal pattern of the daily course of *g*_s_ to be flatter as the artificial CL in climate chambers did. The differences between aboriginal plant species and introduced plants were species-specific in nature and did not allow us to identify any pattern.

## 3. Discussion

In this study, we investigated plant responses to CL provided by artificial light sources and under natural conditions. The plants under study that were exposed to A-CL in the climate chamber where other parameters were held constant exhibited an entire spectrum of responses aimed at protection and adaptation to excessive illumination as were observed in many crop species exposed to artificial CL in growth chambers and PFALs [2,3,18,19,20,21,22,23,24,25,26,27,28,29,30,31,32,33,34,35,36]. These included degreening, a higher Chl *a*/*b* ratio, a lower Chl/carotenoid ratio and lower relative chlorophyll content in the LHCII compared to plants grown under a 16 h photoperiod. Elevated H_2_O_2_ and MDA levels indicated the development of oxidative stress in these plants, and the increased antioxidant capacity was insufficient to cope with the oxidative stress induced by A-CL. The contents of the flavonoids and anthocyanins, which protect plant tissues from excess radiation, were increased. In our experiments, A-CL-treated plants had greater DLI compared to plants grown under A-16 h conditions (21.6 and 14.4 mol/(m^2^ day), correspondingly) and therefore the enhancement of photoprotective mechanisms was rather expected. Some decrease in the *F_v_*/*F_m_* values was also observed, suggesting a possible decrease in photosynthetic activity, although most species showed an increase in LMA values, indicating an accumulation of assimilates in leaves, presumably as a result of their hindered export under CL conditions due to an imbalance between source and sink strengths [3,10,18,35]. Exposure to CL means plants are under constant photon pressure which continuously drives photosynthesis. Constant photosynthesis means the continuous production of photosynthetic products such as soluble sugars and starch [13,36].

The excessive accumulation of carbohydrates can cause damage to chloroplast membranes, which inevitably leads to a downregulation of photosynthesis through a feedback mechanism caused by the over-reduction of the electron transport chain components [37,38]. Many authors also believe that the accumulation of photosynthetic products during CL affects the expression of genes involved in the control of photosynthesis, and ultimately this leads to the complete or partial inhibition of this process [39,40,41,42].

Under natural conditions during the polar day, the leaves of N-CL plants looked healthy, but, later in August, we observed changes in leaves associated with senescence (Figure 6). The symptoms were similar to those observed on A-CL leaves (Figure 5). Color changes in aging leaves were due to the progressive loss in Chl coinciding with the partial retention of carotenoids, the new synthesis of anthocyanins and the formation of dark oxidation products of phenolics. Thus, the leaf photodamages induced by the A-CL were actually symptoms of accelerated leaf senescence. Although the regulation of leaf senescence is age dependent, it is heavily influenced by internal and environmental stimuli. It is thought that one of the processes that is involved in the control and response to senescence regulatory networks is that of the sugar status. A high level of sugar in plants has been shown to both decrease the levels of photosynthetic activity and to initiate the senescence process. Our results are consistent with other research results [10,43,44,45] suggesting that a higher nighttime light intensity without a change in the spectrum can raise the carbohydrate levels in plants and accelerate the leaf senescence that is associated with CL-injury.

Why were the leaves not injured by N-CL during the polar day? Velez-Ramirez et al. [3] rightly noted that in the strict sense, CL does not naturally occur. In natural conditions, even during the polar day, in the absence of complete darkness, light intensity and spectral distribution considerably vary. Irradiance from sunlight changes in a sinusoidal manner during the day (Figure 7). As for the spectral fluctuations, it was shown long ago that they set the circadian clock of arctic birds [46]. A great difference between the spectral quality of artificial and natural light may be one of the reasons why plants did not become injured under N-CL. Artificial lighting produces light spectra with very uneven wavelength distributions, with sharp peaks around the main emission wavelengths and little or no light in between. In contrast, the visible light spectrum of sunlight is much more homogeneous [47]. Earlier experiments have shown that if CL was partially or totally provided by solar light, the injury was reduced or even absent [2,3,18,48,49,50,51,52]. Thus, a question arose: is the CL-induced injury caused by the continuity of light itself or by an interaction between the photoperiod and light spectral distribution?

We add to this that temperature fluctuations under CL can also set the circadian clock in plants under artificial CL [2,3,18,24,25,36,44,53]. As is known, in natural conditions, the temperature fluctuates during the day/night cycle, whereas in most experiments in climate chambers, the temperature is maintained at a constant level. Thus, the results of the CL treatments when other environmental parameters are held constant should be interpreted and given proper care.

It was shown by many authors that plants grown in controlled environments can show significantly different phenotypes from plants of the same genotype grown in more natural conditions or in the field [14,15,54,55,56]. However, the lack of information does not allow us to clearly judge which of the parameters that distinguish whether the natural or artificial environments have the greatest impact on plant metabolism and growth [47]. A unique series of experiments was carried out in the 1930s–1960s by the founders and first researchers who worked for the Polar-Alpine Botanical Garden (KPABG, Kirovsk, Russia, 67°38′ N). They studied responses to CL during the long polar day in indigenous plants and plants that were introduced from other parts of the world. Non-native plants were brought over for agricultural, medicinal and ornamental purposes. In order to exclude the influence of other environmental factors, plants were tested under either the natural day length (24 h) or the shortened day length (8, 10, 12 or 14 h). Daylight hours were shortened by shading using specially designed photoperiodic cabins that moved along rails. The cabin area was 6 m^2^ and the height was 1.2 m. As many as 37 species of trees and shrubs, and 21 species of herbaceous plants were tested. The effects of the long summer daylight on transpiration, photosynthesis, chemical composition, and mineral uptake, as well as the rhythms of growth and development, were investigated and reported. The reports on the experiments conducted have been preserved and are currently available in the archives of the Kola Research Center of the Russian Academy of Sciences. We did not find any mention of leaf injuries due to CL in them, although a slight change in color was sometimes noted. It is what we also observed in our research.

It is known that the exposure of plants to CL can significantly increase their photo-oxidative stress [2,3]. ROS are a normal by-product of photosynthesis, but when generated in large quantities during periods of high irradiance or prolonged light exposure (e.g., CL), they can be harmful to plants. In particular, excessive ROS accumulation can cause severe and irreversible DNA damage, leading to cell death [57]. ROS can also be used as a signaling molecule to alert the plant to stressful conditions, such as high or prolonged light. Thus, a normal balance between ROS production and scavenging can maintain homeostasis since ROS accumulation can initiate the expression of genes whose products are involved in cellular detoxification [57]. Plants with naturally higher levels of antioxidants and enzymes that scavenge ROS have been shown to suffer less damage when exposed to long photoperiods, even with CL [29]. Therefore, it is suggested that the ability to eliminate excess ROS may also play an important role in preventing CL damage based on their role in photooxidative stress [58]. In our study, native plants exposed to CL had higher (by 30–80%) content of flavonoids compared to their non-native counterparts. Plants under N-CL generally had more flavonoids than those under N-16 h conditions. Flavonoids are a numerous and widespread group of phenolic compounds in the plant kingdom [59]. Due to their pronounced antioxidant properties, they participate in the protection of plants from oxidative stress caused by adverse environmental influences. By absorbing ultraviolet radiation (330–350 nm) and some visible rays (520–560 nm), flavonoids protect plant tissues from excess radiation. Thus, the obtained experimental data indicate the participation of flavonoids in plant adaptation to CL. Apparently, the role of flavonoids in the mechanism of the photosynthetic apparatus protection from photodamage under CL conditions is greater in native to subarctic plants compared to introduced plants. This supports the hypothesis that the qualitative and quantitative composition of flavonoids accumulated by plants is closely related to the origin and evolution of a particular species. By conducting this comparative analysis, our research offers valuable insights into the optimal lighting conditions necessary for the successful cultivation of plants in CEA. Sinusoidal LED light recipes instead of square wave light regimes may be of interest for optimizing indoor lighting regimes to more closely match the requirements of the crop, in order to maximize photosynthetic efficiency and consequently marketable yield.

## 4. Materials and Methods

### 4.1. Plant Material

We selected three native plant species (*Geranium sylvaticum* L., *Geum rivale* L., *Potentilla erecta* (L.) Raeusch.), which are common in subarctic woods, and three non-native plant species (*Geranium himalayense* Klotzsch, *Geum coccineum* Sibth. and Sm., *Potentilla atrosanguinea* Loddiges ex D. Don) that were introduced in the Polar-Alpine Botanic Garden (KPABG), the Kola Scientific Center, Russian Academy of Sciences in Kirovsk, Russia (67°38′N) (Figure 2) and commonly used in city landscaping in the subarctic region. *Geranium himalayense* and *Potentilla atrosanguinea* are native to the Himalayas. *Geum coccineum* is native to the mountains of the Balkans. They are all herbaceous perennials that belong to Geraniaceae and Rosaceae families. Species selection was done to match congeneric pairs of native and non-native plant species.

### 4.2. Experimental Design

In the natural environment, measurements of physiological and biochemical parameters were carried out at the peak of the polar day (in the end of June), when the plants were illuminated continuously, and in the second week of August, when the day length was about 16 h. The average temperature, effective temperature sum, relative air humidity, and amount of precipitation during the vegetation period, and the vegetation period length in 2021 and 2022 in the Polar-Alpine Botanic Garden are presented in Table 5.

The non-native plants introduced to KPABG grew on plots with soils classified as agrozem alfe–humus-stratified soils (Table 6) [60]. Native species grew on humus-illuvial podzols.

For indoor experiments, five plants of each of the three species pairs were placed into containers with soil and grown for a month in the climate chamber (Vötsch, Balingen, Germany) at a 16 h photoperiod, with a light intensity of 250 μmol m^−2^ s^−1^, a temperature of 23 °C and air humidity of 70 ± 5%. Then, some of the plants were grown for two weeks under the same conditions, but with a 24 h photoperiod. Light was provided by cool white, fluorescent tubes supplemented with red-enhanced fluorescent tubes. The PPFD value was measured using an LI-250 A Light Meter (Li-COR Biosciences, Lincoln, NE, USA).

Thus, the experimental groups included 3 species pairs exposed to (1) a natural 16 h photoperiod (N-16 h), (2) a natural CL (N-CL), (3) an artificial 16 h photoperiod (A-16 h) and (4) an artificial CL (A-CL).

### 4.3. Leaf Mass Per Area

Five fully expanded leaves of five plants from each experimental group were sampled. Eight discs were cut from each leaf with an 8 mm diameter cork borer. The dry weight of the discs was determined after their drying to a constant weight at 105 °C. The values of leaf mass per area (LMA) were calculated as the ratio of a dry mass of the lamina discs to their area.

### 4.4. Chlorophyll Fluorescence Measurements

The chlorophyll fluorescence parameters of the plants were measured using a Pulse Amplitude Modulation Fluorometer (MINI-PAM, Heinz Walz, Germany). The values of the potential quantum yield of photochemical activity of PSII (F_v_/F_m_) were determined after the leaves were dark-adapted for 30 min with leaf clips.

### 4.5. Stomatal Conductance Measurements

The stomatal conductance (*g_s_*) was measured using a steady state leaf porometer (SC1, Decagon Devices Inc., Pullman, WA, USA). The diurnal course of *g_s_* was measured every 3 h for 3 days, from one of the topmost fully expanded leaves. The leaves used for measurements were tagged for repeated measurements. All the measurements were done on the abaxial side of the leaves.

### 4.6. Photosynthetic Pigment Analysis

The content of chlorophyll a and b and the carotenoids was measured in 96% ethanol extracts with a SF2000 spectrophotometer (Spectrum, St. Petersburg, Russia) and calculated according to the known formulas [61]. The percentage of Chl in the light-harvesting complex II (LHCII) was calculated by accepting that almost all chloropyll b is in LHCII and that the ratio of chlorophyll a/b in LHCII is 1.2 [62].

A SPAD-502 Plus chlorophyll meter (Konica Minolta Optics, Tokyo, Japan) was used to determine the chlorophyll content index (CCI).

### 4.7. Anthocyanins and Flavonoids Content

Anthocyanins were extracted from leaves, according to Kang et al. [63]. Fresh samples (0.1 g) were homogenized in 4 mL of 95% ethanol-1.5 N HCl-(85:15, v:v). After overnight extraction at 4 °C in darkness, each sample was centrifuged at 10,000× *g* for 5 min. The absorbance of the supernatant was measured at 530 nm (peak of absorption of anthocyanin) and 657 nm (peak of absorption of chlorophyll degradation products). The results were plotted as a difference in absorption at 530 and 657 nm relative to tissue fresh weight (∆A∙g^−1^ FW), and the formula ∆A = A530 − 1/4 A657 was used to deduct the absorbance contributed by chlorophyll and its degradation products in the extract [64].

The relative amounts of flavonoids were measured spectrophotometrically [65,66,67]. The supernatant for anthocyanins was diluted 10 times and the absorbance was measured at 300 nm. Flavonoid content in the sample was expressed as absorbance at 300 nm g^−1^ FW.

### 4.8. Malondialdehyde (MDA) Content

The content of malondialdehyde (MDA), the end product of lipid peroxidation, was determined with a standard method based on the reaction of these substances with thiobarbituric acid (TBA) that produces a trimethine complex with an absorption maximum at 532 nm [68]. The value for the nonspecific absorption of each sample at 600 nm was also recorded and subtracted from the absorbance recorded at 532 nm. The concentration of MDA was calculated using an extinction coefficient of 155 mM^−1^ cm^−1^. The lipid peroxidation levels were expressed as micromoles of MDA per gram of FW.

### 4.9. Hydrogen Peroxide Content

Hydrogen peroxide content was determined according to Velikova et al. [69]. Leaf tissues (0.1 g) were homogenized in an ice bath with 2 mL 0.1% (*w*/*v*) TCA. The homogenate was centrifuged at 12,000× *g* for 15 min at 4 °C and 0.5 mL of supernatant was added to 0.5 mL potassium phosphate buffer (pH 7.0) and 1 mL 1 M KI. The absorbance of the supernatant was measured at 390 nm. The content of H_2_O_2_ was calculated by comparison with a standard calibration curve and expressed in μmol g^−1^ FW.

### 4.10. Data Analysis

Five plants per treatment were employed in the experiment. The tables and figures show mean values and standard errors. Significant differences between the means were revealed at *p* < 0.05 using the least significant difference test.

## 5. Conclusions

This research presents a novel approach by examining the specific effects of different lighting environments on plants of different origin. While previous studies have addressed aspects of the responses of important crops to long photoperiods including CL under controlled climate conditions with artificial lighting, there was a lack of comprehensive research comparing natural and artificial lighting. A comprehensive comparison between natural light and artificial lighting sources have shown that plants lack specific mechanisms of tolerance to CL. Protective responses in this case are non-specific and induced by developing photo-oxidative stress. Under constant environmental conditions, artificial CL causes photodamage to leaves due to oxidative stress, the main cause of which is circadian asynchrony, i.e., a mismatch between the internal (endogenous) rhythms of the organism and external light–dark cycles [8].

In nature, during the polar day, plants avoid photodamage because, unlike under artificial conditions, endogenous rhythms are maintained by daily fluctuations in other environmental factors (light intensity, spectral distribution and temperature), which can probably obviously also act as pacemakers. Among various non-specific defense mechanisms, plants actively use flavonoids to neutralize the increased amount of ROS generated by light. At the same time, the role of flavonoids in the photoprotection mechanism is obviously higher in local subarctic plants than in non-native ones. This not only indicates their special role under CL conditions, but also confirms the hypothesis that the qualitative and quantitative composition of flavonoids accumulated by plants is closely related to both the origin of a particular species and its evolution, during which those adaptive mechanisms that were distinguished by their simplicity, reliability and efficiency were selected and consolidated.

## Figures and Tables

**Figure 1 plants-13-02742-f001:**
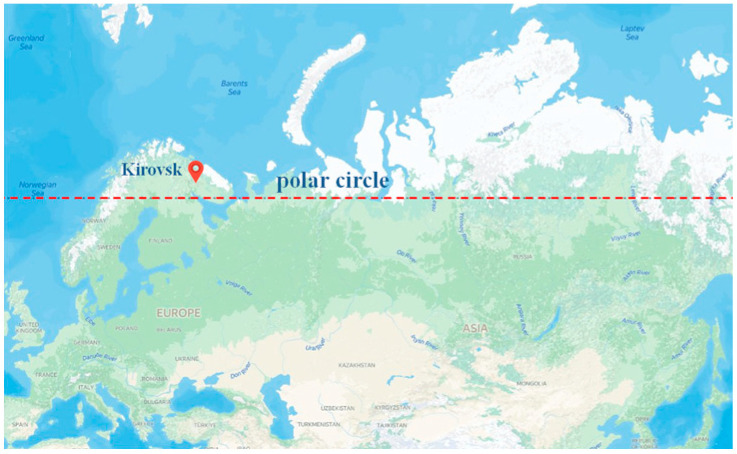
Location of the study site. Polar-Alpine Botanic Garden (KPABG), Kirovsk, Russia (67°38′ N).

**Figure 2 plants-13-02742-f002:**
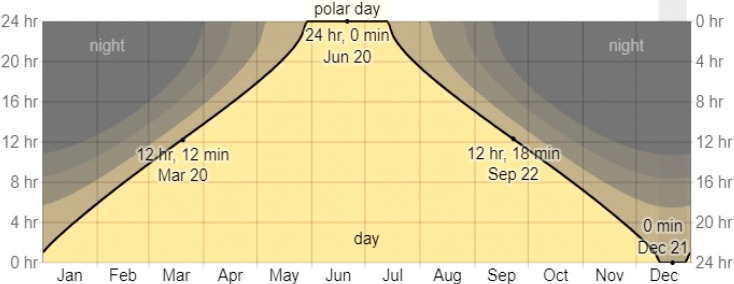
Hours of daylight and twilight in Kirovsk (67°38′ N) © WeatherSpark.com. The number of hours during which the sun is visible (black line). From bottom (most yellow) to top (most gray) the color bands indicate full daylight, twilight (civil, nautical and astronomical) and full night.

**Figure 3 plants-13-02742-f003:**
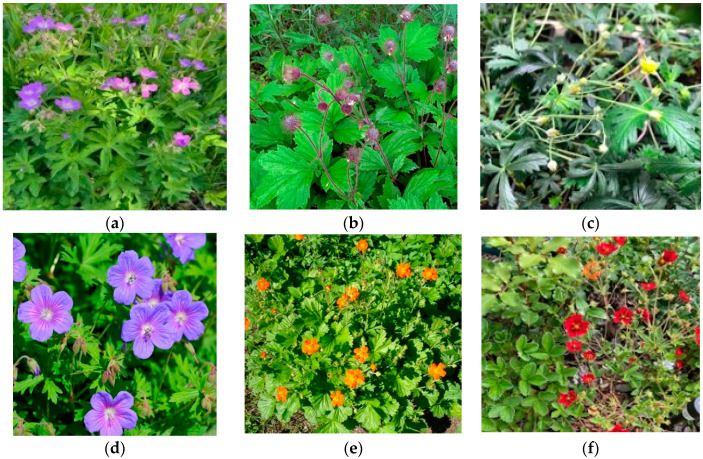
Plants under study. Native-to-subarctic plant species: (**a**) *Geranium sylvaticum* L., (**b**) *Geum rivale* L., (**c**) *Potentilla erecta* (L.) Raeusch.; non-native plant species: (**d**) *Geranium himalayense* Klotzsch, (**e**) *Geum coccineum* Sibth. and Sm., and (**f**) *Potentilla atrosanguinea* Loddiges ex D. Don.

**Figure 4 plants-13-02742-f004:**
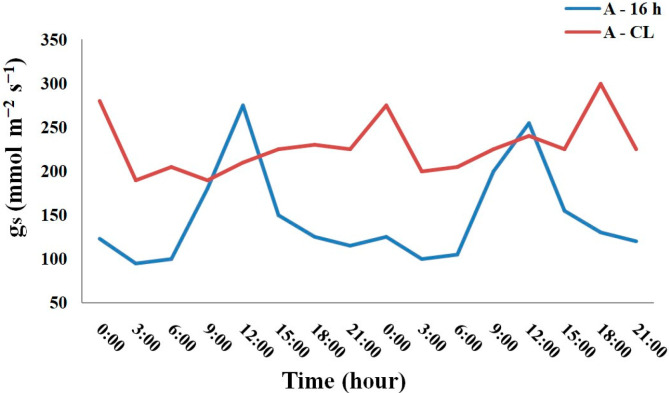
The daily course of stomatal conductance of A-16 and A-CL leaves of *Potentilla atrosanguinea*.

**Figure 5 plants-13-02742-f005:**
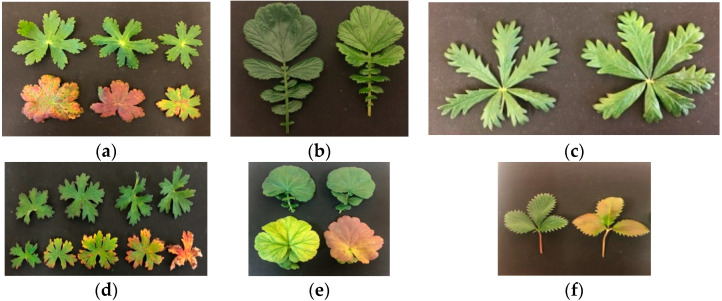
Leaves of plants exposed to A-16 (top on (**a**,**c**–**e**) and left on (**b**,**f**)) and A-CL (bottom on (**a**,**c**–**e**) and right on (**b**,**f**)) for 2 weeks. (**a**) *Geranium sylvaticum*, (**b**) *Geum rivale*, (**c**) *Potentilla erecta*, (**d**) *Geranium himalayense*, (**e**) *Geum coccineum* and (**f**) *Potentilla atrosanguinea*.

**Figure 6 plants-13-02742-f006:**
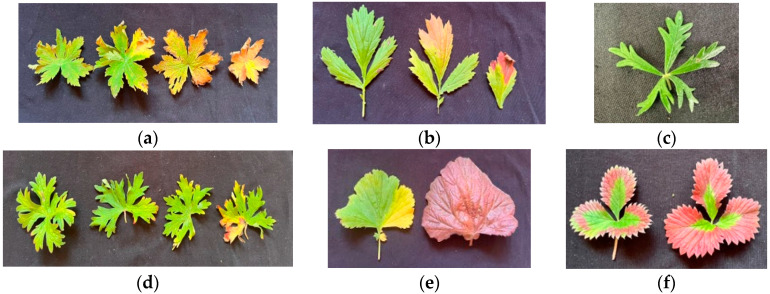
Leaf color pattern in aging N-16 h leaves. (**a**) *Geranium sylvaticum*, (**b**) *Geum rivale*, (**c**) *Potentilla erecta*, (**d**) *Geranium himalayense*, (**e**) *Geum coccineum* and (**f**) *Potentilla atrosanguinea*.

**Figure 7 plants-13-02742-f007:**
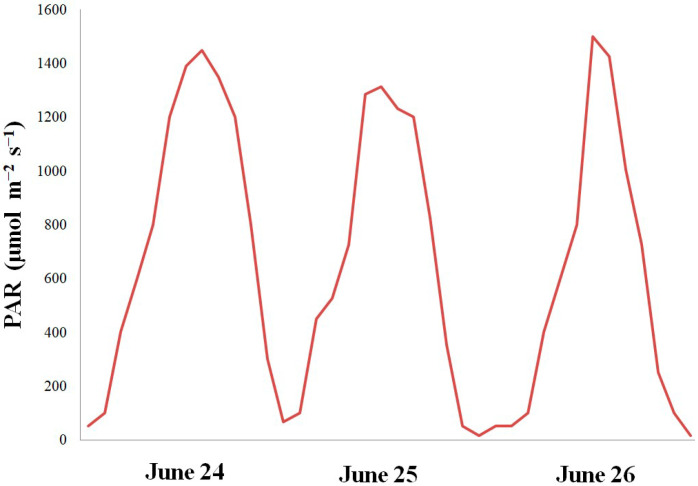
Photosynthesis active radiation (PAR) measured during the polar day on 24–26 June 2024 at Kirovsk, Russia (67°38′ N).

**Table 1 plants-13-02742-t001:** Day length in Khibiny Mountains (67°44′ N); hour.

Days	May	June	July	August	September
1–5	17–42	23–07	24–00	18–41	14–38
6–10	18–24	24–00	23–59	17–57	14–00
11–15	19–08	24–00	22–44	17–18	13–22
16–20	19–53	24–00	21–27	16–40	12–42
21–25	20–46	24–00	20–29	16–02	12–04
26–30	21–52	24–00	19–35	15–09	11–28
Average by month	19–38	24–00	22–02	17–00	13–02

**Table 2 plants-13-02742-t002:** Physiological and biochemical parameters of native and non-native plant species exposed for 2 weeks to artificial lighting with 16 h (A-16 h) or 24 h (A-CL) photoperiods.

Parameter	Photoperiod	*Geranium* *sylvaticum*	*Geranium* *himalayense*	*Geum* *rivale*	*Geum* *coccineum*	*Potentilla* *erecta*	*Potentilla* *atrosanguinea*
LMA, mg/cm^2^	A-16 h	3.52 b	3.00 b	4.78 b	4.63 b	4.94 b	5.25 b
	A-CL	5.37 a	4.98 a	6.87 a	7.3 a	5.29 a	7.96 a
*F_v_*/*F_m_*	A-16 h	0.766 a	0.778 a	0.783 a	0.782 a	0.823 a	0.804 a
	A-CL	0.691 b	0.755 b	0.773 a	0.753 b	0.806 b	0.741 b
Chl, SPAD	A-16 h	28.3 a	37.9 a	44.6 a	40.2 a	50.4 a	49.6 a
	A-CL	26.0 a	33.8 a	43.9 a	22.5 b	49.8 a	37.2 b
Chl *a* + *b*, mg/g DW	A-16 h	3.60 a	3.87 a	8.21 a	7.96 a	8.98 a	7.11 a
	A-CL	2.64 b	3.14 a	5.12 b	2.39 b	8.13 a	3.32 b
Chl *a*/*b*	A-16 h	2.3 b	2.9 a	2.5 a	2.4 a	2.0 a	3.15 b
	A-CL	2.6 a	3.0 a	2.6 a	2.6 a	2.2 a	3.34 a
Carotenoids, mg/g DW	A-16 h	0.51 a	0.68 a	1.0 a	0.98 a	0.72 a	1.02 a
	A-CL	0.49 a	0.69 a	0.7 a	0.35 b	0.56 a	0.61 b
Chl/Carotenoids	A-16 h	7.1 a	5.7 a	8.6 a	8.1 a	12.5 a	7.0 a
	A-CL	5.4 b	4.6 a	6.8 b	6.8 b	14.5 a	5.0 b
LHCII, %	A-16 h	69.9 a	56.4 a	64.0 a	67.9 a	79.3 a	54.4 a
	A-CL	60.4 b	54.8 a	51.6 b	64.7 a	81.7 a	48.4 b
Anthocyanins, mg/mg FW	A-16 h	0.80 b	1.98 b	0.98 b	1.06 b	0.59 b	0.78 b
	A-CL	4.23 a	3.94 a	1.95 a	3.73 a	0.70 a	6.23 a
Flavonoids, mg/mg FW	A-16 h	271.60 b	404.3 b	191.2 b	189.9 b	85.0 b	205.1 b
	A-CL	497.40 a	632.5 a	243.1 a	233.7 a	140.5 a	215.9 a
MDA, μmol/g FW	A-16 h	53.9 b	51.4 b	45.6 b	93.1 b	52.9 b	45.4 b
	A-CL	79.0 a	76.4 a	88.0 a	175.3 a	67.3 a	65.9 a
H_2_O_2_, μmol/g FW	A-16 h	1.36 b	1.10 b	0.78 b	1.47 b	1.06 b	0.89 b
	A-CL	2.58 a	1.85 a	1.86 a	2.02 a	1.47 a	1.44 a

Different letters for each plant species indicate significant differences between the mean values at *p* < 0.05.

**Table 3 plants-13-02742-t003:** Daily variation (maximum/minimum, mmol/m^2^ s) of stomatal conductance (*g*_s_) in native and non-native plant species exposed for 2 weeks to artificial lighting with 16 h (A-16 h) or 24 h (A-CL) photoperiods and exposed to a natural 16 h photoperiod (N-16 h) in the second week of August or natural continuous lighting (N-CL) at the end of June in the Polar-Alpine Botanic Garden (KPABG, 67°38′ N).

Plant Name	Maximum/Minimum Daily *g*_s_, mmol/(m^2^ s)			
	A-16 h	A-CL	N-16 h	N-CL
*Geranium sylvaticum*	229/142	312/275	996/330	686/244
*Geranium himlayense*	363/90	297/188	711/267	645/248
*Geum rivale*	440/250	395/320	845/304	776/286
*Geum coccineum*	215/98	123/101	477/183	450/197
*Potentilla erecta*	507/160	521/223	812/195	790/217
*Potentilla atrosanguinea*	342/114	297/2 60	995/302	806/124

**Table 4 plants-13-02742-t004:** Physiological and biochemical parameters of native and non-native plant species exposed to natural 16 h photoperiod (N-16 h) in the second week of August or natural continuous lighting (N-CL) in the end of June in Polar-Alpine Botanic Garden (KPABG, 67°38′ N).

Parameter	Photoperiod	*Geranium* *sylvaticum*	*Geranium* *himalayense*	*Geum* *rivale*	*Geum* *coccineum*	*Potentilla* *erecta*	*Potentilla* *atrosanguinea*
LMA, mg/cm^2^	N-16 h	3.83 b	3.77 b	3.23 a	4.82 b	3.95 b	5.57 b
	N-CL	4.74 a	4.23 a	3.47 a	7.50 a	5.99 a	6.81 a
*F_v_*/*F_m_*	N-16 h	0.805 a	0.821 a	0.820 a	0.813 a	0.820 a	0.808 a
	N-CL	0.795 a	0.802 b	0.818 a	0.794 b	0.821 a	0.800 a
Chl, SPAD	N-16 h	22.8 a	41.3 a	43.3 a	40.2 a	40.8 a	45.1 a
	N-CL	20.6 a	33.4 b	38.3 a	26.5 b	45.1 a	38.9 b
Chl *a + b*, mg/g DW	N-16 h	3.21 a	5.24 a	3.03 a	6.43 a	7.28 a	4.51 a
	N-CL	2.71 b	2.26 b	1.98 b	2.27 b	6.47 b	1.24 b
Chl *a*/*b*	N-16 h	2.6 b	3.0 b	3.0 a	1.9 b	1.8 b	2.9 b
	N-CL	3.4 a	3.6 a	3.4 a	2.5 a	2.5 a	3.7 a
Carotenoids, mg/g DW	N-16 h	0.52 a	0.68 a	0.50 a	0.57 a	0.82 a	0.77 b
	N-CL	0.44 b	0.41 b	0.37 a	0.28 b	0.62 b	0.66 a
Chl/Carotenoids	N-16 h	6.2 a	7.8 a	6.1 a	11.3 a	8.9 a	5.9 a
	N-CL	6.3 a	5.5 b	5.4 a	8.1 b	10.4 a	1.9 b
LHCII, %	N-16 h	61.2 a	6.1 a	55.0 a	77.9 a	78.5 a	56.8 a
	N-CL	50.5 b	47.8 b	49.7 a	51.2 b	61.0 b	42.6 b
Anthocyanins, mg/mg FW	N-16 h	1.18 a	0.84 a	0.87 a	0.82 b	1.17 a	0.98 a
	N-CL	1.04 a	0.95 a	1.02 a	1.17 a	1.07 a	0.76 b
Flavonoids, mg/mg FW	N-16 h	254.7 a	354.1 a	279.8 b	334.8 a	82.1 b	131.4 b
	N-CL	277.5 a	283.6 b	343.1 a	351.3 a	139.0 a	215.7 a
MDA, μmol/g FW	N-16 h	57.1 b	37.0 b	101.2 b	61.3 a	52.0 a	38.1 a
	N-CL	69.7 a	45.0 a	120.9 a	72.8 b	55.7 a	40.3 a
H_2_O_2_, μmol/g FW	N-16 h	4.48 b	2.10 b	1.9 b	1.62 b	1.82 b	1.76 a
	N-CL	6.42 a	4.01 a	2.8 a	2.54 a	2.22 a	1.82 a

Different letters for each plant species indicate significant differences between the mean values at *p* < 0.05.

**Table 5 plants-13-02742-t005:** Average temperature, effective temperature sum, relative air humidity, amount of precipitation during the vegetation period and vegetation period length in 2021 and 2022 in Polar-Alpine Botanic Garden (KPABG, 67°38′ N).

Parameter		Year	
		2021	2022
Average temperature, °C	June	13.3	11.9
	July	14.8	16.5
	August	11.4	14.1
Effective temperature sum (base 5 °C)	June	From May 30 to Sept 11 401.0	From May 28 to Sept 29 356.8
	July	457.4	508.6
	August	350.9	436.9
Effective temperature sum (base 10 °C)	June	From June 2 to Aug 28 375.8	From May 29 to Aug 28 282.5
	July	407.3	499.6
	August	265.2	415.8
Relative air humidity, %	June	7.6	79.9
	July	76.9	85.0
	August	88.3	85.9
Amount of precipitation during the vegetation period, mm		375	353
Vegetation period length, days		66	75

**Table 6 plants-13-02742-t006:** Physico-chemical parameters of soils (mg/kg) in Polar-Alpine Botanic Garden (KPABG, 67°38′ N).

Horizon, Depth (sm)	pH	Humus, (Cx1.724), %	TiO_2_	Fe_2_O_3_	CaO	Al_2_O_3_	SiO_2_	P_2_O_5_	K_2_O	MgO	MnO
Humus-illuvial podzols (Native plants)											
O 0–10	5.2	49.0	1.17	4.5	3.3	7.2	61.0	1.43	1.43	0.73	0.08
E 10–20	4.9	2.5	1.66	2.6	2.9	15.2	78/0	0.26	2.15	0.67	0.06
BHF 20–25	5.6	16.5	0.68	8.7	1.1	21.3	61.1	0.38	0.61	0.68	0.05
BHF 25–30	5.3	23.4	0.58	9.7	0.9	18.0	59.8	0.54	0.58	0.78	0.03
BC 30–50	5.9	7.5	0.85	5.7	1.6	26.9	64.5	0.4	2.02	0.66	0.15
Agrozem alfe–humus-stratified soils (Non-native plants)											
Prh 0–15	6.7	4.7	1.87	9.6	2.6	28.0	64.0	0.59	2.22	1.11	0.39
Ccr 15–26	6.5	2.4	1.76	9.1	1.9	32.3	67.0	0.27	2.23	0.95	0.39
[E] 26–30	6.1	4.1	1.96	6.1	3.5	18.3	67.6	0.19	2/01	1.26	0.12
[BH] 30–38	5.9	23.2	0.78	8.4	1.2	20.5	59.7	0.50	0.6	0.79	0.05
[BHF] 30–50	5.9	23.3	0.53	6.7	1.1	21.2	60.6	0.51	0.58	0.69	0.06

## Data Availability

The data presented in this study are available on request from the corresponding author.

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
