# Peer review of "Response of Native and Non-Native Subarctic Plant Species to Continuous Illumination by Natural and Artificial Light"

_plants, 2024, doi:10.3390/plants13192742_

Round 1

Reviewer 1 Report

Comments and Suggestions for Authors

There are many experimental details that are missing, making evaluation difficult.

For example, we are not given air temperature conditions for the outdoor experiments, for either growth period.  How do they compare with the constant 23 C of the indoor experiments?  The indoor photon flux density of 250 is very low compared with the outdoor conditions.  How were the plants grown in the outdoor experiments: in pots? fertilizer? age? duration of exposure?  Why was stomatal conductance measured only one leaf surface?  In herbs, both surfaces usually have significant conductance.  It is stated that stomatal conductance was measured every 3 hours.  Was this also true when the light duration was only 16 hours, both indoors and outdoors? 

Author Response

We thank the reviewer for the valuable comments and revommendations.

Here is the list of improvements made accordingg to the recommendation.

  1. For example, we are not given air temperature conditions for the outdoor experiments, for either growth period.  How do they compare with the constant 23 C of the indoor experiments?           Reply - We added Table 5 that shows data on average temperatures, the sum of effective temperatures (base 5 and 10°C), the amount of precipitation, and the duration of the vegetation period.
  2. The indoor photon flux density of 250 is very low compared with the outdoor conditions. Reply:  Yes, we agree that this PPFD is lower than under sunlight, but even at this level of illumination, the plants in the climate chambers developed signs of photodamage. It is logical to assume that higher PPFD would accelerate the damage process.

  3. How were the plants grown in the outdoor experiments: in pots? fertilizer? age? duration of exposure? Reply: In fact, we did not conduct experiments as such in the open air, but carried out measurements on plants, and the plants grew a) in the soil on sites prepared for introduced plants and b) in the forest on the territory of the botanical garden. Photoperiodic exposure was natural (see Table 1 and Fig. 2). The plants were at the age corresponding to the phenological phases of June, July, and August.
  4. Why was stomatal conductance measured only one leaf surface?  In herbs, both surfaces usually have significant conductance. Reply:: Indeed, both leaf sides (adaxial and abaxial) have stomatal conductance. In our study the absolute value of stomatal conductance was not as important as the daily dynamics. The aim was to see if it becomes flattened under the constant lighting or not. Thus, we measured stomatal conductance at the back of the leaf because it provides a more accurate representation of the stomatal behavior and serves to minimize external factors that may affect the measurements. 
  5. It is stated that stomatal conductance was measured every 3 hours.  Was this also true when the light duration was only 16 hours, both indoors and outdoors? Reply: Yes, in all cases, stomatal conductance measurements were carried out every 3 hours for several days in each experimental variant.

Reviewer 2 Report

Comments and Suggestions for Authors

The manuscript by Shibaeva et al. investigated the response of native and non-native to subarctic plant species to continuous illumination by natural and artificial light, the application of CL for plant cultivation is interest to both plant scientists and commercial plant producers. However, this manuscript has some serious drawbacks especially in the Methods part.

1, Line15-18, the experiment design should be briefly introduced in the Abstract part.

2, Figure 4, only gs of Potentilla atrosanguinea was shown in this figure, but  how about the gs variations of other plant species?

3, Line 347-349, what were the temperatures in the end of June and in the second decade of August? Would the temperature cause impact on those plants?

4, In 4.2. Experimental Design section, what was the age of plants used for the N-CL and A-Cl treatments? 

Were the age of plants used for the N-CL treatment same with those used for A-CL treatments?

Besides, I noticed that the treatment period of A-16 (one month) was different from that of A-24 (two weeks), what was the consideration? If the plants treated in A-16h and A-24h were at different growth stages, would the growth stage itself influence the plant physiology and biochemical parameters?

5, P351, more details about the soil should be described. For example, what type of soil did the authors use in this experiment? How about the physical and chemical properties? Moreover, what was the soil type and characteristics in the nature of this region?

6, Line 353, the authors described the environmental factors in the climate chamber, however, how about the environmental factors in the natural environment? Were they different significantly? If so, how to exclude the difference caused by the impact of environmental factors on the plants in N-CL and A-Cl treatments? And it was hard to determine whether the difference between plant responses in N-CL and A-Cl treatments was caused by the light source.

Author Response

We thank the reviewer for the valuable comments and reccomendations.

1, Line15-18, the experiment design should be briefly introduced in the Abstract part.

Reply: Thank you, done. Red-marked.

2, Figure 4, only gs of Potentilla at sanguinea was shown in this figure, but  how about the gs variations of other plant species?

Reply: We indicated that we provide Figure 4 as an example, and other species demonstrated a similar pattern of dependence, i.e. under artificial lighting continuous illumination made the sinusoidal daily fluctuations in gs to be flatter. It seemed to us unnecessary to provide six identical figures, but this can be done if the reviewer insists. Added to the text: In all species A-CL made the sinusoidal pattern of the daily course of gs to be flatter, i.e. the damping of oscillation was observed.

3, Line 347-349, what were the temperatures in the end of June and in the second decade of August? Would the temperature cause impact on those plants?

Reply: We added Table 5 that shows data on average temperatures, the sum of effective temperatures (base 5 and 10°C), the amount of precipitation, and the duration of the vegetation period. In the conclusion we indicate that the fluctuating temperature in the daily cycle in the natural environment is presumably one of the factors that entrains circadian clock and serves as a pacemaker for endogenous rhythms, which could be one of the reasons for the absence of photodamage to plants under continuous lighting in the natural environment.

4, In 4.2. Experimental Design section, what was the age of plants used for the N-CL and A-Cl treatments? Were the age of plants used for the N-CL treatment same with those used for A-CL treatments? Besides, I noticed that the treatment period of A-16 (one month) was different from that of A-24 (two weeks), what was the consideration? If the plants treated in A-16h and A-24h were at different growth stages, would the growth stage itself influence the plant physiology and biochemical parameters?

Reply: The plants were transferred to the climate chambers in June, i.e. the plant age corresponded to the plants in nature. The duration of exposure to photoperiods of 16 h and 24 h was the same - 2 weeks. Before this, all the plants were under conditions of a 16 h photoperiod for a month to adapt after transplantation.

5,Line  P351, more details about the soil should be described. For example, what type of soil did the authors use in this experiment? How about the physical and chemical properties? Moreover, what was the soil type and characteristics in the nature of this region?

Reply: Thank you. The data on soil property are given in Table 6. 

6, Line 353, the authors described the environmental factors in the climate chamber, however, how about the environmental factors in the natural environment? Were they different significantly? If so, how to exclude the difference caused by the impact of environmental factors on the plants in N-CL and A-Cl treatments? And it was hard to determine whether the difference between plant responses in N-CL and A-Cl treatments was caused by the light source.

Reply: The main conclusion of the article is that it is the environmental factors that fluctuate in the daily cycle in nature and remain constant in the climate chambers, that determine the plant response to continuous illumination. Consequently, as in the natural environment continuous illumination is accompanied by fluctuations in light intensity, its spectral composition, temperature and humidity, circadian asynchrony does not occur in plants and, consequently, photodamage to leaves does not occur.

Round 2

Reviewer 2 Report

Comments and Suggestions for Authors

I suggest acceptance of the present form, since the authors have addressed all my questions.